# Don't Lose the Thread: Empowering Long-Horizon LLM Agents with Cognitive Resource Self-Allocation

## Abstract

Agents powered by large language models (LLMs) have demonstrated remarkable progress in solving complex reasoning tasks. However, LLM agents often falter on long-horizon tasks due to cognitive overload, as their working memory becomes cluttered with expanding and irrelevant information, which dilutes their attention and hinders effective planning and reasoning. To mitigate this challenge, we introduce **CO**gnitive **R**esource Self-**AL**location (**CORAL**), a novel reasoning paradigm that empowers agents to proactively optimize their context. Implemented as an agent-callable working memory management toolset, CORAL allows an agent to maintain crucial checkpoints of its progress within its working memory and adaptively initiate a new problem-solving episode by purging cluttered working memory and resuming its reasoning from the most recent checkpoint, effectively reallocating agentic cognitive resources by implicitly sharpening their attention on the checkpoints. We further enhance the agent's checkpoint capabilities using a Multi-episode Agentic Reinforced Policy Optimization algorithm. On several long-horizon task benchmarks, CORAL significantly outperforms standard LLM agent methods. Notably, analysis of the LLMs' attention distribution reveals that CORAL substantially optimizes agentic RL dynamics, which in turn ensures agents maintain a focused cognitive resource allocation, thereby continuously amplifying performance gains.

## 1 Introduction

Recentlt, LLM-driven agents represent a powerful paradigm that extends the capabilities of Large Language Models (LLMs) through the integration of external tools (OpenAI, 2025b; Gemini, 2025; Liu et al., 2025; Li et al., 2025b), substantially outperforming methods reliant on single-turn inference. To address long-horizon tasks, these agents operate on a THOUGHT-ACTION-OBSERVATION cycle (Yao et al., 2022), engaging in multiple cycles of planning, environmental interaction, and reasoning (Erdogan et al., 2025; Qiao et al., 2024; Huang et al., 2024). A critical challenge arises as each cycle populates LLM agents' context with verbose environmental feedback and a history of failed attempts (Wu et al., 2025b; Shinn et al., 2023). This escalating contextual noise progressively degrades the model's planning and reasoning faculties (Yang et al., 2025), a phenomenon comparable to the cognitive overload that impairs human problem-solving when working memory becomes saturated.

Current paradigms for context optimization in LLM agents seek to prevent this contextual bloat by pruning messages or distilling salient information. The activation of these methods is generally governed by rule-based heuristics, such as fixed intervals (Zhou et al., 2025b; Yu et al., 2025a) or the imminent saturation of the context window (Wu et al., 2025c). The underlying mechanism for optimization typically involves either truncating the context directly (Luo et al., 2025) or utilizing external models to achieve compression (Wu et al., 2025c).

Fundamentally, this redundant context is a direct consequence of the agent's imperfect planning and reasoning. Suboptimal tool use generates a high volume of irrelevant environmental feedback (Wang et al., 2025), which in turn clutters the context and further degrades the agent's reasoning, creating a vicious cycle. To address this, one line of research employs agentic reinforcement learning (RL),

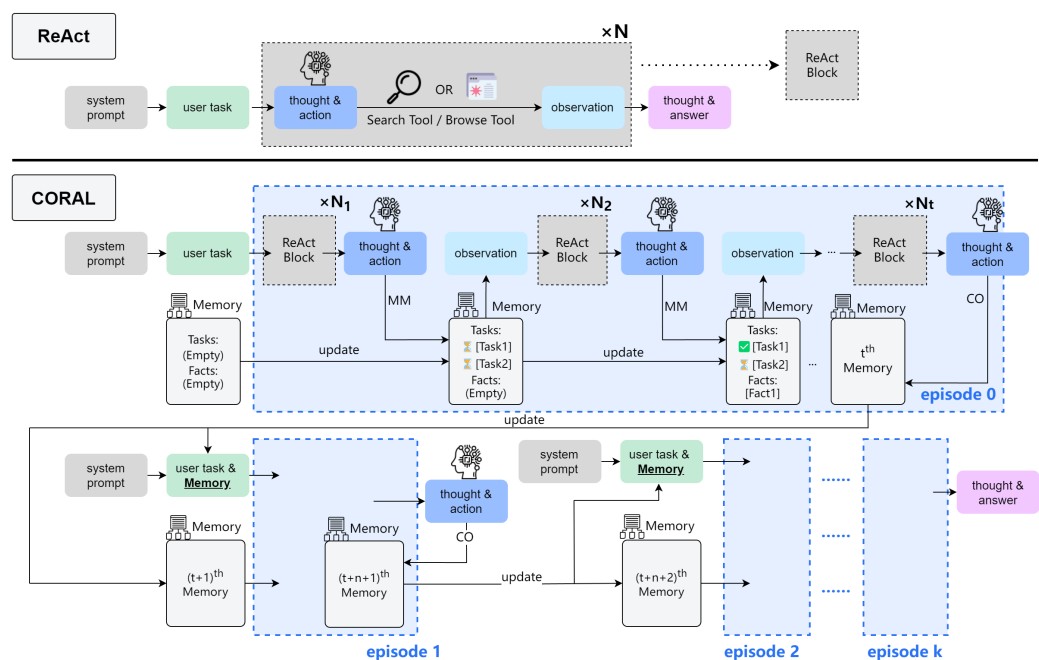

Figure 1: Comparison of the ReAct and CORAL frameworks. CORAL enhances the standard ReAct loop by incorporating two key components: a Memory Management (MM) tool and a Context Optimization (CO) tool. The memory is designed to store two categories of information: task progress and verified facts. The CO tool periodically resets the model's context, which segments a complete trajectory into a series of independent units termed *episodes*.

using algorithms such as GRPO (Shao et al., 2024b) and DAPO (Yu et al., 2025b) with carefully designed reward functions to optimize the agent's tool-use policy (Qian et al., 2025; Zhang et al., 2025; Jin et al., 2025). However, in long-horizon tasks involving multi-step interactions, these RL methods face significant challenges with reward sparsity. Relying solely on final outcomes makes it difficult to assign credit to intermediate actions, leading to unstable and inefficient training dynamics. While methods like estimated step-level credit assignment can mitigate this (Feng et al., 2025; Xia et al., 2025; Chandrahasan et al., 2025), proactive context optimization presents a powerful, orthogonal method of improving RL training dynamics (Kimi, 2025; Wu et al., 2025c).

To address these challenges, we introduce the **CO**gnitive **R**esource Self-**AL**location (**CORAL**) framework. CORAL extends existing agentic architectures with a callable toolset for working memory management, empowering an agent to dynamically optimize its context and sustain high-level planning and reasoning throughout long-horizon tasks. Specifically, the agent can autonomously invoke memory tools to create checkpoints of its progress and verified facts. The periodic insertion of these checkpoints along the task trajectory systematically refocuses the agent's attention on its most current state, preventing cognitive resources from being squandered on obsolete information, such as prior environmental feedback or failed attempts. This process facilitates an implicit yet effective self-allocation of cognitive resources. Furthermore, CORAL allows the agent to adaptively initiate new problem-solving episodes by purging its working memory and resuming its reasoning from the latest checkpoint. We initially enhance the crucial checkpointing ability, the capacity to accurately distill key task information through Supervised Fine-Tuning (SFT).

To further enable the model to discover optimal checkpointing strategies without additional trajectory data, we introduce a Multi-episode Agentic Reinforced Policy Optimization (Multi-episode ARPO) algorithm. This approach not only refines the agent's checkpointing policy but also significantly improves the overall agentic RL dynamics. We validate CORAL's effectiveness on the GAIA benchmark, where it substantially outperforms existing LLM agent methods on long-horizon tasks (Levels 2 and 3). Analysis of the action-level attention distribution reveals the source of this suc-

cess: CORAL enables the agent to efficiently allocate its cognitive resources throughout the entire reasoning process.

In summary, the key contributions of this work are as follows:

- We introduce **CO**gnitive **R**esource Self-**AL**location (**CORAL**), a framework that empowers agents to manage their working memory through a callable toolset. By dynamically optimizing its own context, an agent using CORAL can maintain robust planning and reasoning capabilities on long-horizon tasks.

- We use Supervised Fine-Tuning (SFT) to instill core checkpointing skills and then leverage a multi-episode agentic reinforced policy optimization (**Multi-episode ARPO**) algorithm to allow the agent to discover optimal checkpointing strategies.

- On the GAIA benchmark, CORAL significantly outperforms existing approaches on complex long-horizon tasks (Level-2 and Level-3). An analysis of action-level attention distributions confirms that CORAL's success stems from its ability to effectively allocate the agent's cognitive resources during reasoning.

## 2 PRELIMINARIES

### 2.1 PROBLEM FORMULATION

We consider a general large language model (LLM)-based agent. Upon receiving a problem specification, the agent is capable of interacting with its environment and executing a sequence of reasoning and action steps to progressively derive a solution. Following the ReAct (Yao et al., 2022) framework, these steps can be formalized as iterations of *Thought-Action-Observation*. Specifically, given a question $q \in p(Q)$, the LLM agent $\pi_\theta$ at time step $t$ generates *Thought* $r_t \sim \pi_\theta(\cdot|c_t)$ and a textual *Action* $a_t \sim \pi_\theta(\cdot|c_t, r_t)$. The $c_t$ denotes the context in the time step $t$: $c_t = (q, r_1, a_1, o_1, ..., r_{t-1}, a_{t-1}, o_{t-1})$. Then the environment gives the feedback as the *Observation* $o_t$. The loop ends when the agent solves the question or reaches the max steps. Therefore, the final episode with $M$ steps can be defined as:

$$e_{\text{terminated}} = (q, r_1, a_1, o_1, ..., r_M, a_M, o_M) \tag{1}$$

$$e_{\text{completed}} = (q, r_1, a_1, o_1, ..., r_M) \tag{2}$$

Note that in the completed episode, the *Thought* in the final round ($r_M$) contains the answer to the question, and the episode stops immediately.

### 2.2 WEB SEARCH AGENTIC TOOL DESIGN

At each time step $t$, the LLM-based agent generates a textual *Action* $a_t \in \mathcal{A}$, where $\mathcal{A}$ denotes the predefined action space. In this work, we focus on an LLM-based tool-use agent, in which the action space $\mathcal{A}$ comprises a set of specialized tool-use commands and interaction primitives that the agent can execute to accomplish complex tasks. To operationalize this action space, we design two purpose-built tools that collectively support web search and webpage browse. These tools are described as follows:

- **Web Search.** Enables the agent to issue multiple search queries in parallel via a search engine, retrieve and format the results, and present them in a structured manner.

- **Web Browse.** Allows the agent to intelligently retrieve and analyze content from specified web pages according to a user-defined goal, extract relevant information, summarize key findings, and identify useful external links for further exploration.

## 3 COGNITIVE RESOURCE SELF-ALLOCATION (CORAL)

Inspired from cognitive resource theory, which posits that effective problem-solving in humans relies on the strategic management of finite cognitive resources like attention and working memory, we draw a parallel to the operational challenges faced by LLM agents. The agent's context serves as its working memory. On long-horizon tasks, this "memory" becomes progressively cluttered with

intermediate steps (thoughts, actions, and observations), leading to cognitive overload. To address this, we introduce **CO**gnitive **R**esource Self-**AL**location (**CORAL**), a paradigm that empowers the agent to proactively manage its own cognitive load. CORAL allows the agent to mimic the human process of consolidating progress and refocusing attention by creating checkpoints and purging irrelevant context.

### 3.1 WORKING MEMORY MANAGEMENT TOOLSET

We operationalize this paradigm through the following working memory management toolset:

- **Memory Management.** Assists the agent in managing its working memory by adding or removing knowledge units, thereby retaining essential information across context resets while discarding outdated or irrelevant data to maintain clarity and task continuity.

- **Context Optimization.** Performs a hard reset of the conversational context to mitigate token bloat and sustain performance. It clears all conversational history except for essential components—such as working memory, system prompt, and the original user request—ensuring that critical information is preserved while resetting the token count and removing accumulated tool outputs.

  Specifically, in the time step $t$, the context is $(q, r_1, a_1, o_1, ..., r_{t-1}, a_{t-1}, o_{t-1})$, the LLM-based agent call the Context Optimization tool, *i.e.* $a_t = a_{CO}$, the tool response $o_t$ will be the next round's context $c$. Therefore, in the next round, the episode will begin like $(c, r_1, a_1, o_1, ...)$.

**Multi-episode trajectory.** The context optimization tool, as described, performs a hard reset of the conversational context. This reset operation effectively segments what would otherwise be a single continuous reasoning process into multiple shorter episodes, each starting with a refreshed context while retaining only essential information. To capture this behavior, we extend the single-episode formulation to a *multi-episode trajectory*. We assume that there are $N$ episodes in total, then the i-th episode (with $M_i$ iterations) can be formulated as:

$$e_i = \begin{cases} (c_i, r_{i,1}, a_{i,1}, o_{i,1}, ..., r_{i,M_i}, a_{i,M_i}, o_{i,M_i}), & i < N \\ (c_i, r_{i,1}, a_{i,1}, o_{i,1}, ..., r_{i,M_i}), & i = N \end{cases} \tag{3}$$

$$\text{where} \quad c_i = \begin{cases} q, & i = 1 \\ o_{i-1,M_{i-1}}, & i > 1 \end{cases} \tag{4}$$

Notice that in the final episode, the thought of the last iteration $r_{N,M_N}$ contains the answer, then the episode ends immediately. $c_i$ is the initial context of each episode. In the first episode, it is the question $q \in p(Q)$. While in the following episodes, it is the optimized context $o_{i-1,M_{i-1}}$ from the last *Context Optimization* tool. Then a complete trajectory with $N$ episodes can be defined as:

$$\mathcal{T} = (e_1, e_2, ..., e_N) \tag{5}$$

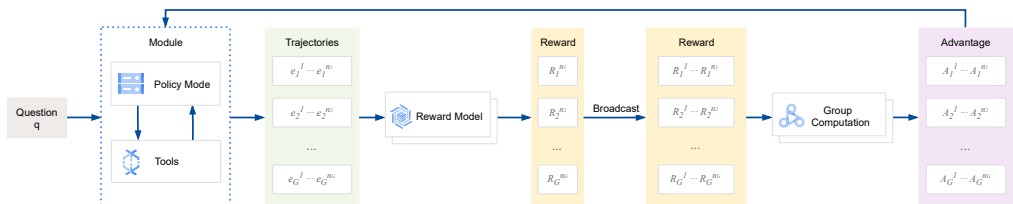

Figure 2: Multi-episode DAPO. The reward of a multi-episode trajectory is computed using the last episode, which contains the answer. Then the reward is broadcasted to all previous episodes in the same trajectory.

### 3.2 CORAL FRAMEWORK

We propose the COgnitive Resource Self-ALlocation (CORAL) framework, a novel reasoning paradigm designed to empower LLM agents to overcome cognitive overload in long-horizon tasks.

The framework is built on the principle that agents should be able to proactively manage their own context, much like humans manage their working memory. As illustrated in Figure 1, we implement this capability by augmenting the traditional ReAct framework with a specialized working memory management toolset. This toolset, comprising two distinct tools, provides the agent with the explicit mechanisms needed to self-regulate its cognitive load. Specifically, the Memory Management tool enables the agent to consolidate its progress and focus on planning, while the Context Optimization tool acts as a reset mechanism, allowing it to strategically purge irrelevant information from its context.

### 3.3 FURTHER ENHANCEMENT METHODS

While the CORAL framework can be implemented in a prompting-only fashion, we explore dedicated training methods to further enhance its capabilities. **Behavior Cloning.** To endow the agent with basic function call ability, we apply behavior cloning through supervised fine-tuning (SFT) on curated, high-quality trajectories. From Equation 5 we know that a trajectory is consist of multiple context independent episodes, therefore, we split the trajectory into episodes, and fine-tune the model using batches of episodes. For each episode described in Equation 3 and Equation 4, we compute the loss using the following loss function:

$$L = -\frac{1}{\sum_{i=1}^{|e|} \mathbb{I}(x_i \neq o)} \sum_{i=1}^{|e|} \mathbb{I}(x_i \neq o) \cdot \log \pi_\theta(x_i \mid x_{<i}) \tag{6}$$

where $\mathbb{I}(\cdot)$ is the indicator function. Here $\mathbb{I}(x_i \neq o)$ masks out the loss from observation tokens, ensuring the loss is computed over the agent's own generated outputs, such as its reasoning steps (thoughts) and function calls (actions). By doing so, we only supervise the model on the behaviors it is expected to learn, rather than penalizing it for failing to predict external information from the environment.

**Multi-episode Agentic Reinforced Policy Optimization.** The classic DAPO optimization objective in Agent Reinforcement Learning (Wu et al., 2025a):

$$\mathcal{J}_{\text{DAPO}}(\theta) = \quad \mathbb{E}_{(q,a)\sim\mathcal{D},\{o_i\}_{i=1}^G \sim \pi_{\theta_{\text{old}}}(\cdot|context)}$$

$$\left[ \frac{1}{\sum_{i=1}^G |o_i|} \sum_{i=1}^G \sum_{t=1}^{|o_i|} \min\left( r_{i,t}(\theta)\hat{A}_{i,t}, \text{ clip}\left(r_{i,t}(\theta), 1 - \varepsilon_{\text{low}}, 1 + \varepsilon_{\text{high}}\right)\hat{A}_{i,t}\right)\right] \tag{7}$$

$$\text{s.t.} \quad 0 < \left| \left\{ o_i \mid \texttt{is\_equivalent}(a, o_i) \right\} \right| < G,$$

where

$$r_{i,t}(\theta) = \frac{\pi_\theta(o_{i,t} \mid q, o_{i,<t})}{\pi_{\theta_{\text{old}}}(o_{i,t} \mid q, o_{i,<t})}, \quad \hat{A}_{i,t} = \frac{R_i - \text{mean}(\{R_i\}_{i=1}^G)}{\text{std}(\{R_i\}_{i=1}^G)}. \tag{8}$$

Noted that agentic execution $o_i$ refers solely to the tokens generated by models, excluding any tool responses. It means the optimization is applied only to the model-generated tokens.

In this work, the trajectory consists of multiple episodes, as mentioned in Equation 5. We further extend the Agentic DAPO algorithm to handle multi-episode trajectories by treating each episode as a separate optimization unit while maintaining trajectory-level coherence. Figure 2 illustrates our main idea. For a multi-episode trajectory $\mathcal{T}_i = (e_i^1, e_i^2, ..., e_i^N)$, we use the last episode to compute the reward. And all previous episodes in the same trajectory share this reward: $R_i^j = R_i^{n_i}$ for $1 \leq j < n_i$. Then all episodes participate in the group computation to get an advantage.

**Reward Design.** We design a simple reward function that consist of format reward $R_i^{format}$ and answer reward $R_i^{answer}$. The format reward verifies whether the whole trajectory follows the pre-defined format, and all the tool call in the *json* format is valid. The answer reward uses a LLM as a judge to determine whether the final answer is correct.

$$R_i = R_i^{format} \times R_i^{answer} \tag{9}$$

Table 1: Main results on GAIA. We **boldface** the best performance and underline the second best performance. Models with size 7 or 8B and models larger than 32B are marked separately. "-" means results that are not reported.

| Model | Level 1 | Level 2 | Level 3 | Average |
|---|---|---|---|---|
| DIRECT INFERENCE | | | | |
| GPT-4o | 23.1 | 15.4 | 8.3 | 17.5 |
| DeepSeek-R1 | 43.6 | 26.9 | 8.3 | 31.1 |
| Claude-4.0-Sonnet | 38.5 | 36.5 | 8.3 | 34.0 |
| AGENTIC INFERENCE | | | | |
| R1-Searcher-7B | 28.2 | 19.2 | 8.3 | 20.4 |
| WebDancer-7B | 41.0 | 30.7 | 0.0 | 31.0 |
| WebSailor-7B | - | - | - | 37.9 |
| CK-Pro-8B | **56.4** | 42.3 | 8.3 | **43.7** |
| WebDancer-32B | 46.1 | 44.2 | 8.3 | 40.7 |
| WebThinker-32B-RL | 56.4 | 50.0 | 16.7 | 48.5 |
| WebSailor-72B | - | - | - | 55.4 |
| WebShaper-72B | - | - | - | 60.1 |
| OpenAI DR | 74.3 | 69.1 | 47.6 | **67.4** |
| CONTEXT OPTIM | | | | |
| ReAct 🅰 | - | - | - | 60.0 |
| +HARD OPTIM | - | - | - | 66.0 |
| ReAct 🔷 | 33.3 | 11.5 | 8.3 | 19.4 |
| +HARD OPTIM | 28.2 | 19.2 | 0.0 | 20.4 |
| +SFT | 41.0 | 40.4 | 11.1 | 37.2 |
| +RL | 41.0 | **44.2** | **25.0** | 40.9 |

## 4 EXPERIMENTS

### 4.1 EXPERIMENTAL SETTINGS

**Baselines.** We compare our method with against three representative paradigms.

- **Direct Inference**: GPT-4.1 (OpenAI, 2025a), DeepSeek-R1 (Guo et al., 2025), Claude-4.0-Sonnet
- **Agentic Inference**: R1-Searcher (Song et al., 2025), WebDancer (Wu et al., 2025a), Web-Thinker (Li et al., 2025a), WebSailor, WebShaper, OpenAI Deep research (OpenAI, 2025b)
- **ReAct.** Classic ReAct diagram using web search and web browse tools.

**Benchmarks.** We use GAIA (Mialon et al., 2023) as the evaluation benchmark. We follow existing works by using the 103-sample text-only validation subset. Questions are categorized into three difficulty levels, with Level 3 representing the most challenging long-horizon tasks requiring extensive reasoning chains.

**Dataset.** We follow the data construction pipeline of WebShaper (Tao et al., 2025) to construct high quality questions with controllable difficulty. We use commercial models to synthesize the interaction trajectories. Ultimately, this process yielded a dataset of 1115 trajectories, of which approximately 55% successfully lead to the correct answer.

### 4.2 OVERALL PERFORMANCE

Table 1 resents our main experimental results. CORAL demonstrates substantial improvements over existing methods, particularly excelling on the most challenging long-horizon tasks (Level 2

and Level 3) that require extended reasoning chains. When applied to a powerful proprietary model (Claude-4-Sonnet), our prompting-only CORAL achieves an average score of 66.0, comparable with OpenAI DR (67.4). This highlights the effectiveness of our approach even when enhancing already capable models.

However, when applied to Qwen3-8B, the prompting-only CORAL shows only marginal improvements. This can be attributed to a discrepancy between the sophistication of CORAL's working memory management tools and the limited agentic capabilities of the base model. The model frequently fails to adhere to the required format or makes errors during tool calling, which negates the potential benefits of the framework.

This phenomenon can be mitigated through behavior cloning, i.e., by performing Supervised Fine-Tuning (SFT) on high-quality trajectories. Remarkably, our experiments demonstrate that SFT on a small dataset of just 1115 trajectories is sufficient for the model to master this operational pattern and achieve superior performance on GAIA. This is achieved even though 45% of the trajectories in the training data culminate in an incorrect final answer, suggesting the model is effectively learning the reasoning process itself. The subsequent application of Reinforcement Learning (RL) further enhances performance, with the advantages being most pronounced on long-horizon tasks (Level 2 and Level 3).

### 4.3 ABLATION STUDY

**Does the trajectory with wrong answer degrade model's performance?** To investigate this question, we fine-tuned a base model exclusively on trajectories from our dataset that resulted in correct answers. This model achieved a score of 31.1% on the GAIA text-only subset, a result substantially lower than that of our model fine-tuned on the complete dataset (which includes both correct and incorrect trajectories).

This finding indicates that including trajectories with incorrect answers is not only harmless but is, in fact, beneficial. This aligns with our hypothesis that the primary goal of Supervised Fine-Tuning (SFT) is to "clone behavior", where the value gained from learning a high-quality reasoning process outweighs the negative signal of an incorrect final answer. Therefore, even high-quality reasoning paths that conclude with an incorrect answer can positively contribute to the model's overall reasoning capabilities. However, whether this conclusion remains valid when the dataset is scaled up significantly requires further investigation.

### 4.4 ATTENTION ANALYSIS

CORAL has shown significant improvements over baseline methods, particularly on challenging long-horizon tasks requiring extended reasoning chains.

In this section, we move beyond a macro-level evaluation of the CORAL framework's performance to a micro-level analysis of the underlying mechanisms driving its success. The central hypothesis is that the CORAL framework implicitly facilitates a more efficient and effective reallocation of model's cognitive resources. This analysis uses attention mechanisms as a lens to investigate how the model learns to prioritize critical information and steer its problem-solving trajectory.

**Receiver heads.** Previous work (Bogdan et al., 2025) in attention analysis has identified "important sentences" that receive heightened attention from downstream sentences, a phenomenon known as attention aggregation. Inspired by this, we also try to find important parts in the context that might get higher attention values and thus have a greater impact on the model's behavior. In our multi-turn conversation setting, we shift the unit of analysis from tokens or sentences to messages, aiming to discover which messages are more important. Following (Bogdan et al., 2025), we refer to attention heads that narrow attention toward specific messages as "receiver heads". We first identify the receiver heads (details in Appendix A.2), then analysis the attention distribution through these heads.

**Case study: Sharpening attention on checkpoints.** In Figure 3, we show a case of message-level attention from the base model and fine-tuned model. The attention map clearly shows that, after fine-tuning, the model pays more attention to previous checkpoint (memory management tool

response) when calling memory management tool, while other part of the context show a relatively lower attention value. We also find that,

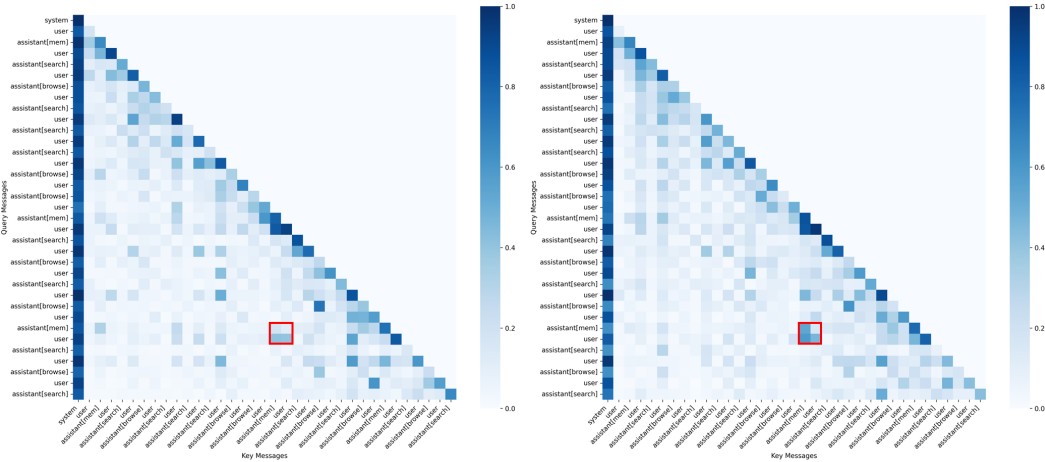

Figure 3: Comparison of attention at a checkpoint between the base model and the fine-tuned model in CORAL diagram. **Left:** Message-level attention map from the Qwen3-8B base model. **Right:** Message-level attention map from our fine-tuned Qwen3-8B. **Red box:** Attention corresponding to two consecutive memory management tool calls.

## 5 RELATED WORK

**Reinforcement learning for LLM agents.** Reinforcement learning (RL) is a crucial methodology for empowering Large Language Model (LLM) agents to operate effectively within dynamic and open-ended environments. Compared to supervised fine-tuning which relies on pre-collected expert data, RL-based methods allow agents to learn directly from their interactions with an environment. The application of RL to LLM agents has evolved significantly over time. Initial efforts utilized classical algorithms like DQN for training agents in text-based games (Narasimhan et al., 2015). Subsequently, more advanced value-based methods, such as PPO (Schulman et al., 2017) and GRPO (Shao et al., 2024a), were employed in a broader array of interactive settings, including embodied AI tasks like ALFWorld (Shridhar et al., 2021), information seeking tasks (Mialon et al., 2023; Wei et al., 2025; Zhou et al., 2025a; Xbench-Team, 2025), and strategic card games (Brockman et al., 2016).

**Context Engineering in LLM Agents.** Managing context effectively is a critical challenge in developing LLM-based agents, particularly as these systems become more sophisticated and operate over extended interactions. Recent research has explored various approaches to address the limitations of context windows and maintain relevant information throughout agent execution. One prominent approach involves breaking down complex tasks into smaller, manageable subtasks to better utilize limited context windows (Luo et al., 2025; Schroeder et al., 2024). Another line of research focuses on employ context compression after each function call (Zhou et al., 2025b). While this approach can effectively manage context size, it may suffer from information loss and difficulties in maintaining high-level planning coherence across extended agent interactions. Some systems have begun to incrementally read context by splitting it into chunks (Yu et al., 2025a). However, they have only considered scenarios with fixed contexts, while dynamic contexts involving function calling remain unexplored.

## 6 CONCLUSION

In conclusion, we address the critical challenge of contextual bloat in LLM-driven agents, where the accumulation of environmental feedback and intermediate reasoning steps degrades performance on long-horizon tasks. We introduce the COgnitive Resource Self-ALlocation (CORAL) framework, a novel paradigm that empowers agents with a callable toolset to actively manage their own working

memory. By learning to create checkpoints and strategically reset its context, an agent equipped with CORAL can mitigate cognitive overload and sustain high-level reasoning throughout a task. Our two-stage training approach, which combines Supervised Fine-Tuning to instill core skills with a novel Multi-episode Agentic Reinforced Policy Optimization (Multi-episode ARPO) algorithm, enables the agent to discover effective, adaptive memory management policies. On the challenging Level 2 and Level 3 tasks of the GAIA benchmark, CORAL substantially outperforms existing methods. Our analysis of action-level attention distributions confirms that this performance gain is directly attributable to the agent's improved ability to allocate its cognitive resources, focusing on salient information while discarding obsolete context.

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

## A  APPENDIX

### A.1  THE USE OF LARGE LANGUAGE MODELS (LLMS)

We used Large Language Models (LLMs) to aid in polishing the language of this manuscript. Their role was confined to improving grammar, clarity, and sentence structure. The intellectual content, including all ideas and findings, is entirely the work of the human authors, who reviewed and approved the final text.

### A.2  THE IDENTIFICATION OF RECEIVE HEADS

Receive heads refers to the attention heads which consistently narrow attention toward specific messages. Following (Bogdan et al., 2025), we plot the vertical attention scores for each message by the 32 different heads in 36 different layers. From Figure 4, We find that in later layers (layer 35) shows a clear difference in attention values between different attention heads. In this case, the receive heads are head 9 and head 22 in layer 35.

We take a look at these two head's message level attention map, find that these two attention really show a relatively high attention value (see Figure 5 and Figure 6). And narrow attention toward specific messages, such as the 6th message in head 22.

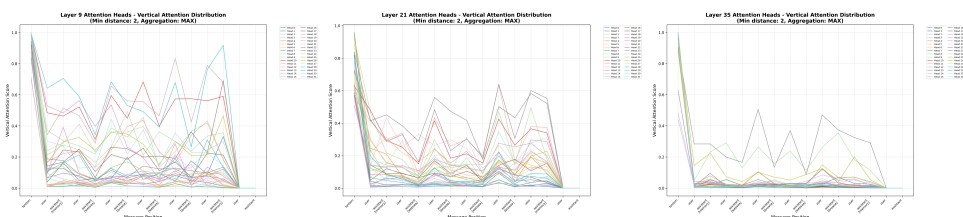

Figure 4: Vertical attention scores for each message by 32 different heads in layer 9, 21, 35 respectively. The backbone of the tested model is Qwen3-8B.

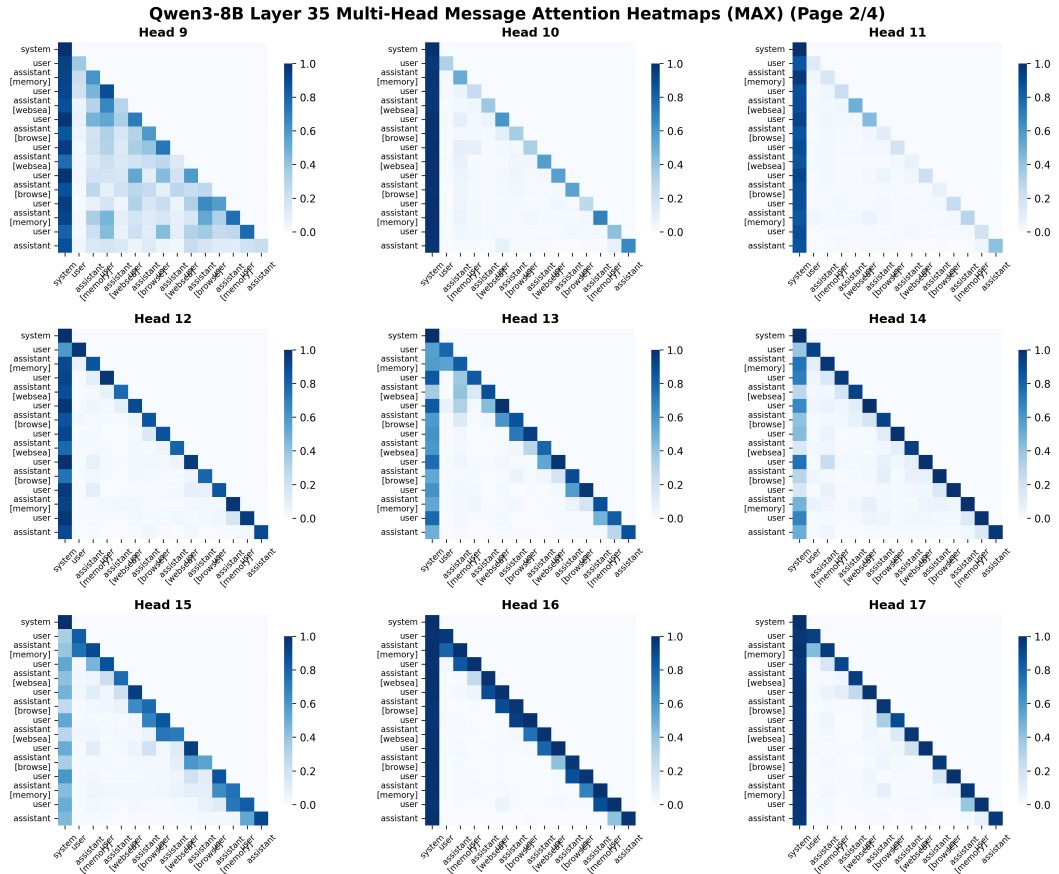

Figure 5: Message level attention map for head 9 layer 35 and its neighbors.

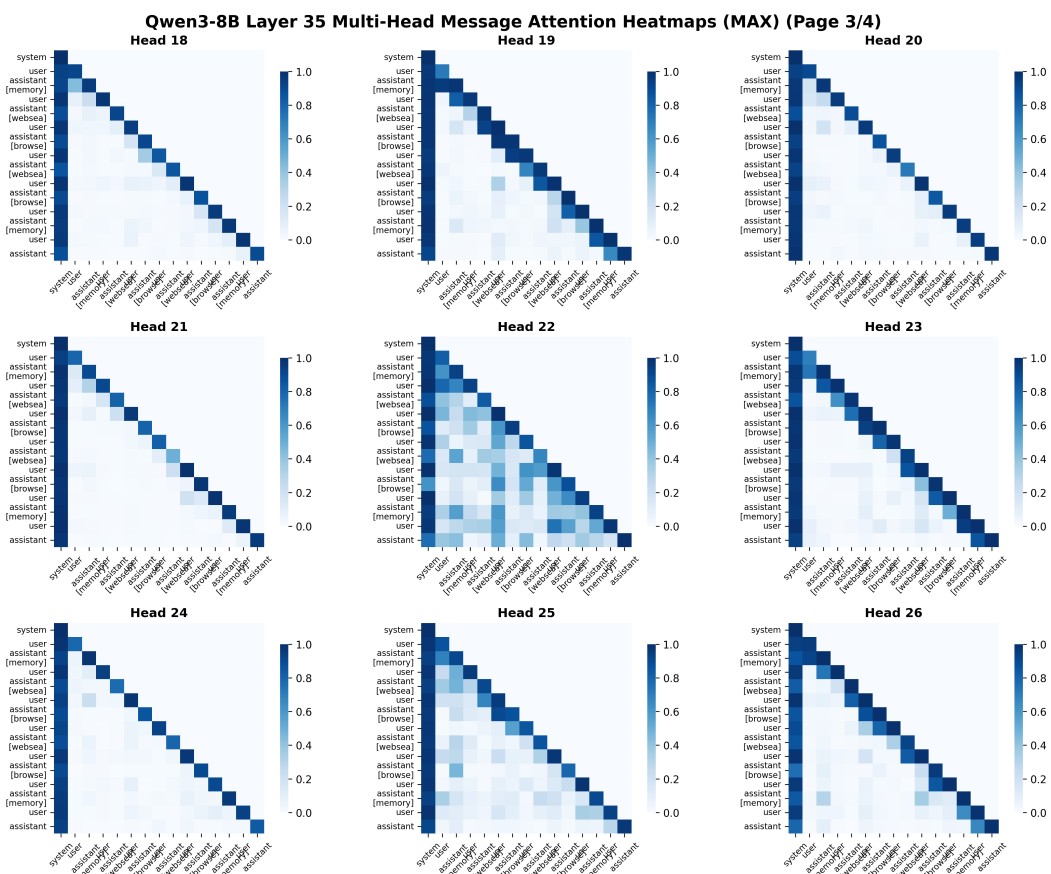

Figure 6: Message level attention map for head 22 layer 35 and its neighbors.

