# OpenReview forum: "Don’t Lose the Thread: Empowering Long-Horizon LLM Agents with Cognitive Resource Self-Allocation"
_ICLR.cc/2026/Conference — ICLR 2026 Conference Withdrawn Submission_

### Official Review · Reviewer_z3nr · 2025-10-29

**Soundness:** 3
**Presentation:** 3
**Contribution:** 3
**Rating:** 4
**Confidence:** 3

**Summary:**

The authors propose COgnitive Resource Self-ALlocation (CORAL), a reasoning paradigm that allows LLM agents to dynamically manage and optimize their own context during multi-step tasks. CORAL equips agents with a working-memory management toolset, consisting of a memory management tool and a context optimization tool. These mechanisms allow agents to periodically “reboot” their reasoning—focusing attention on recent progress and freeing cognitive resources. For training enhancements, authors use supervised fine-tuning (SFT) to teach checkpointing behavior using high-quality trajectories, and present multi-episode agentic reinforced policy optimization (Multi-episode ARPO) to refine this behavior by treating multi-episode trajectories as cohesive training units, propagating rewards from final outcomes to earlier checkpoints. The authors show that, on the GAIA benchmark (a test for reasoning and research agents), CORAL significantly outperforms existing frameworks such as ReAct, WebThinker, and DeepSeek-R1—especially on long-horizon tasks (Levels 2–3).

**Strengths:**

1. The paper is generally well-structured and readable.
2. It reframes context management as a self-allocation problem, drawing an explicit analogy to human working-memory control — a fresh and underexplored perspective in LLM research. The underlying principle — that LLMs should manage their own cognitive bandwidth — is a conceptual contribution to the future of autonomous reasoning agents.
3. The CORAL framework creatively combines existing elements — the ReAct agent loop, memory tools, and reinforcement learning — into a coherent, self-optimizing reasoning paradigm.

**Weaknesses:**

1. While CORAL is presented as a novel reasoning paradigm, several of its core ideas overlap with existing work on memory-aware and context-optimization agents, such as Thread (Schroeder et al., 2024), MemAgent (Yu et al., 2025), and Mem1 (Zhou et al., 2025).
2. The empirical evaluation relies almost exclusively on the GAIA benchmark, which, while widely used, focuses on text-based research and reasoning tasks with synthetic structure. This narrow scope raises questions about the generality of CORAL’s advantages. Specific issues:
No evidence is provided on multi-modal, embodied, or tool-heavy environments (e.g., ALFWorld, WebArena, or XBench-DeepSearch);
Results use proprietary models (Claude-4, GPT-4o) for prompting-based experiments, limiting reproducibility; The paper does not explore context length scaling: it claims to mitigate “cognitive overload” but never tests tasks that explicitly push the context window (e.g., 100K–1M tokens).
3. The paper introduces Multi-episode Agentic Reinforced Policy Optimization (Multi-episode ARPO), which propagates rewards from the final episode backward across all earlier ones. However it lacks comparison to established reinforcement frameworks like DAPO (Yu et al., 2025) and GRPO (Shao et al., 2024) on convergence stability.

**Questions:**

see the weakness section.

---

### Official Review · Reviewer_Nj4W · 2025-10-29

**Soundness:** 2
**Presentation:** 3
**Contribution:** 2
**Rating:** 2
**Confidence:** 3

**Summary:**

This paper presents CORAL, a reasoning framework designed to alleviate cognitive overload in long-horizon LLM agents. The key idea is to grant the agent metacognitive control over its working memory through two callable tools that enable it to create checkpoints of verified facts and task progress, and to periodically reset its context to refocus on essential information. CORAL integrates these mechanisms into the ReAct loop and introduces a multi-episode Agentic Reinforced Policy Optimization (ARPO) algorithm to train memory management policies that optimize checkpointing and reset decisions across multiple reasoning episodes. Experiments on the GAIA benchmark demonstrate substantial gains on long-horizon tasks.

**Strengths:**

* Metacognitive framing: CORAL goes beyond context-window optimization and positions the LLM agent as a self-regulating cognitive system. The notion of "cognitive resource self-allocation" aligns well with human working-memory theory and opens a promising direction for computational metacognition.
* Memory management as actions: Implementing self-regulation through explicit callable tools for memory management and context optimization is elegant and compatible with current agentic frameworks. The separation between context and working memory provides a clear conceptual foundation for scalable reasoning.
* Multi-episode ARPO: The Multi-episode ARPO formulation effectively models credit assignment across episodes, providing a learning mechanism for when to checkpoint or reset. This is a clever extension of PPO-style optimization to agentic reasoning.
* Empirical evidence of attention re-focusing: The attention-map analysis convincingly shows sharper focus on checkpoint tokens after fine-tuning, supporting the cognitive-resource argument.

**Weaknesses:**

* Limited evaluation: The authors used only the GAIA benchmark for evaluation, which limits the generalizability of CORAL’s validity. Would CORAL remain effective in other domains like embodied reasoning, tool-use planning, or knowledge-based QA? Isn't the observed gain specific to GAIA’s web-search and reasoning tasks? I suggest embodied planning e.g. AlfWorld would be a nice domain to test the validity of the proposed framework.
* Ambiguous definition of “verified facts”: The paper states that "the agent can autonomously invoke memory tools to create checkpoints of its progress and verified facts.", but it never defines how "verified facts" are detected and filtered. Are these based on positive observations, confidence thresholds, or LLM judgments? Without a clear criterion for forming verified facts, checkpoint quality and reproducibility are uncertain.
* Limited description of checkpoint summarization: The process of converting ongoing reasoning into concise checkpoints is crucial but underspecified. What summarization prompts or heuristics were used? Different summarization strategies could yield drastically different outcomes.
* Partial reporting of results: Table 1 omits level-wise scores for certain settings (e.g., ReAct + Hard Optimization with Claude). Even if those experiments were limited, aggregated numbers alone obscure performance trends across difficulty levels.

**Questions:**

Please refer to the weaknesses part.

---

### Official Review · Reviewer_uaYK · 2025-11-01

**Soundness:** 2
**Presentation:** 1
**Contribution:** 2
**Rating:** 2
**Confidence:** 4

**Summary:**

This paper introduces CORAL, a agentic method that manages its context while attempting to solve a query. The authors equip the agent with tools for managing its context and optimize it using both SFT and RL. They present results on GAIA, comparing with a range of agent and direct interface baselines, showing that the prompt-based system, that + STF, and that + STF + RL successively improve models.

**Strengths:**

Originality. The context management tool ideas (memory management and context optimization) are to my knowledge new ways of approaching managing decision-making as an agent’s history grows.

Quality. The experiments involve a substantial set of reasonable baselines, all done on a reasonable benchmark, with a useful ablation study.

Clarity. I found that I could generally follow what was going on (with some exceptions, see below).

Significance. The paper demonstrates the efficacy of the various parts of their technique, showing promising performance comparable with strong baselines.

**Weaknesses:**

I found the presentation quite hard to follow in various important places.

1. I couldn’t precisely understand what Memory Management is doing exactly — I have a decent idea but I could not replicate it. What is a “knowledge unit”?
2. I think I know what’s going on with ARPO (episodes share the reward at the end but are optimized separately — but otherwise DAPO?), but several key points for how it is described (”a separate optimization unit while maintaining trajectory-level coherence”, “all episodes participate in the group computation to get an advantage”) are vague when they could be said exactly.
3. The experiments presentation is difficult to follow. That section doesn’t use the method name given in the rest of the paper. One of the critical baselines (indeed the one that outperforms your method overall, CK-Pro-8B) is as far as I can tell never mentioned in the main text. Claude doesn’t have score breakdowns into Level 1, 2, 3, without explanation.

The experimental results seem promising but overall do not beat a baseline (and this isn’t discussed at all, just that the paper’s approach has “superior performance”). I might also suggest, especially your top qwen model does comparably less well (relative to baselines) on Level 1, to do an error analysis on each level.

More minor points: what is o in equation 6? You mean that x_i is masked for any observation, right? And section 3.2 seems largely redundant given 3.1.

**Questions:**

I think all important questions were asked in talking through weaknesses.

---

### Official Review · Reviewer_NPZM · 2025-11-01

**Soundness:** 2
**Presentation:** 1
**Contribution:** 2
**Rating:** 4
**Confidence:** 4

**Summary:**

The paper proposes CORAL, which lets an LLM web agent manage "working memory" via two tools: Memory Management and Context Optimization (CO), where CO performs a hard context reset that preserves essentials (system prompt, original query, working memory) and segments reasoning into episodes. Reset timing is learned: a multi-episode ARPO objective treats each episode as an optimization unit, with a reward that multiplies a format check and an LLM judge answer score. Experiments use the GAIA showed CORAL used purely as a prompting framework on a strong closed model performs competitively with leading systems on a reasoning benchmark. With a smaller open-source base model, adding SFT and RL transforms brittle tool calling into more reliable multi-step behavior and improves performance over a standard ReAct-style baseline. The evaluation, however, is concentrated on a single benchmark and one web environment, and some comparisons leave compute-parity details unspecified.

**Strengths:**

- The proposed mechanism including multi-episode RL setup with a learned context reset is sound. A reset segments trajectories into episodes with clear initial contexts and recurrence (previous CO output becomes the next episode’s seed), which cleanly supports learning when to reset, and might effectively deal with long context issues.

**Weaknesses:**

- The authors claim CORAL "demonstrates substantial improvements", especially on harder levels, and also describes a prompting-only run on a strong closed model that’s "comparable" to a claude. Those narrative claims are present, but some supporting table cells are missing (no per-level breakdowns for key rows), so the evidence is incomplete in places. For example:
  - Table 1 seems incomplete. Several entries are left unreported, and the "+HARD OPTIM" rule-based variant isn't described-so it's hard to judge how learned resets compare against strong heuristics. Also there is no justification ever provided for why these results were excluded.
  - Comparisons span heterogeneous stacks without clear parity on budgets, step limits, and decoding/stopping settings aren’t detailed enough to guarantee apples-to-apples conclusions.

- Although improvements on GAIA would be valid, it would be better to see if this method generalizes across domain.

- It's unclear what the "take-away" should be from Figure 3. The paper shows a single case study attention map and says it focuses more on prior checkpoint messages after fine-tuning, but it doesn’t clearly explain how heads were chosen, how attention was aggregated, or how to read axes and color scales.

**Questions:**

Do you have analysis on when CO is invoked, including the distribution of episode counts, triggers preceding resets, or correlation between reset timing and success?

---

### Note · Authors · 2025-11-20

I have read and agree with the venue's withdrawal policy on behalf of myself and my co-authors.